# Importance Resampling for Off-policy Prediction

**Matthew Schlegel**
University of Alberta
mkschleg@ualberta.ca

**Wesley Chung**
University of Alberta
wchung@ualberta.ca

**Daniel Graves**
Huawei
daniel.graves@huawei.com

**Jian Qian**
University of Alberta
jq1@ulberta.ca

**Martha White**
University of Alberta
whitem@ulberta.ca

## Abstract

Importance sampling (IS) is a common reweighting strategy for off-policy prediction in reinforcement learning. While it is consistent and unbiased, it can result in high variance updates to the weights for the value function. In this work, we explore a resampling strategy as an alternative to reweighting. We propose Importance Resampling (IR) for off-policy prediction, which resamples experience from a replay buffer and applies standard on-policy updates. The approach avoids using importance sampling ratios in the update, instead correcting the distribution before the update. We characterize the bias and consistency of IR, particularly compared to Weighted IS (WIS). We demonstrate in several microworlds that IR has improved sample efficiency and lower variance updates, as compared to IS and several variance-reduced IS strategies, including variants of WIS and V-trace which clips IS ratios. We also provide a demonstration showing IR improves over IS for learning a value function from images in a racing car simulator.

## 1 Introduction

An emerging direction for reinforcement learning systems is to learn many predictions, formalized as value function predictions contingent on many different policies. The idea is that such predictions can provide a powerful abstract model of the world. Some examples of systems that learn many value functions are the Horde architecture composed of General Value Functions (GVFs) [Sutton et al., 2011, Modayil et al., 2014], systems that use options [Sutton et al., 1999, Schaul et al., 2015a], predictive representation approaches [Sutton et al., 2005, Schaul and Ring, 2013, Silver et al., 2017] and systems with auxiliary tasks [Jaderberg et al., 2017]. Off-policy learning is critical for learning many value functions with different policies, because it enables data to be generated from one behavior policy to update the values for each target policy in parallel.

The typical strategy for off-policy learning is to reweight updates using importance sampling (IS). For a given state $s$, with action $a$ selected according to behavior $\mu$, the IS ratio is the ratio between the probability of the action under the target policy $\pi$ and the behavior: $\frac{\pi(a|s)}{\mu(a|s)}$. The update is multiplied by this ratio, adjusting the action probabilities so that the expectation of the update is as if the actions were sampled according to the target policy $\pi$. Though the IS estimator is unbiased and consistent [Kahn and Marshall, 1953, Rubinstein and Kroese, 2016], it can suffer from high or even infinite variance due to large magnitude IS ratios, in theory [Andradottir et al., 1995] and in practice [Precup et al., 2001, Mahmood et al., 2014, 2017].

There have been some attempts to modify off-policy prediction algorithms to mitigate this variance.[1] Weighted IS (WIS) algorithms have been introduced [Precup et al., 2001, Mahmood et al., 2014, Mahmood and Sutton, 2015], which normalize each update by the sample average of the ratios. These algorithms improve learning over standard IS strategies, but are not straightforward to extend to nonlinear function approximation. In the offline setting, a reweighting scheme, called importance sampling with unequal support [Thomas and Brunskill, 2017], was introduced to account for samples where the ratio is zero, in some cases significantly reducing variance. Another strategy is to rescale or truncate the IS ratios, as used by V-trace [Espeholt et al., 2018] for learning value functions and Tree-Backup [Precup et al., 2000], Retrace [Munos et al., 2016] and ABQ [Mahmood et al., 2017] for learning action-values. Truncation of IS-ratios in V-trace can incur significant bias, and this additional truncation parameter needs to be tuned.

An alternative to reweighting updates is to instead correct the distribution before updating the estimator using weighted bootstrap sampling: resampling a new set of data from the previously generated samples [Smith et al., 1992, Arulampalam et al., 2002]. Consider a setting where a buffer of data is stored, generated by a behavior policy. Samples for policy $\pi$ can be obtained by resampling from this buffer, proportionally to $\frac{\pi(a|s)}{\mu(a|s)}$ for state-action pairs $(s, a)$ in the buffer. In the sampling literature, this strategy has been proposed under the name Sampling Importance Resampling (SIR) [Rubin, 1988, Smith et al., 1992, Gordon et al., 1993], and has been particularly successful for Sequential Monte Carlo sampling [Gordon et al., 1993, Skare et al., 2003]. Such resampling strategies have also been popular in classification, with over-sampling or under-sampling typically being preferred to weighted (cost-sensitive) updates [Lopez et al., 2013].

A resampling strategy has several potential benefits for off-policy prediction.[2] Resampling could even have larger benefits for learning approaches, as compared to averaging or numerical integration problems, because updates accumulate in the weight vector and change the optimization trajectory of the weights. For example, very large importance sampling ratios could destabilize the weights. This problem does not occur for resampling, as instead the same transition will be resampled multiple times, spreading out a large magnitude update across multiple updates. On the other extreme, with small ratios, IS will waste updates on transitions with very small IS ratios. By correcting the distribution before updating, standard on-policy updates can be applied. The magnitude of the updates vary less—because updates are not multiplied by very small or very large importance sampling ratios—potentially reducing variance of stochastic updates and simplifying learning rate selection. We hypothesize that resampling (a) learns in a fewer number of updates to the weights, because it focuses computation on samples that are likely under the target policy and (b) is less sensitive to learning parameters and target and behavior policy specification.

In this work, we investigate the use of resampling for online off-policy prediction for known, unchanging target and behavior policies. We first introduce Importance Resampling (IR), which samples transitions from a buffer of (recent) transitions according to IS ratios. These sampled transitions are then used for on-policy updates. We show that IR has the same bias as WIS, and that it can be made unbiased and consistent with the inclusion of a batch correction term—even under a sliding window buffer of experience. We provide additional theoretical results characterizing when we might expect the variance to be lower for IR than IS. We then empirically investigate IR on three microworlds and a racing car simulator, learning from images, highlighting that (a) IR is less sensitive to learning rate than IS and V-trace (IS with clipping) and (b) IR converges more quickly in terms of the number of updates.

## 2 Background

We consider the problem of learning General Value Functions (GVFs) [Sutton et al., 2011]. The agent interacts in an environment defined by a set of states $\mathcal{S}$, a set of actions $\mathcal{A}$ and Markov transition dynamics, with probability $\mathrm{P}(s'|s, a)$ of transitions to state $s'$ when taking action $a$ in state $s$. A GVF is defined for policy $\pi : \mathcal{S} \times \mathcal{A} \to [0, 1]$, cumulant $c : \mathcal{S} \times \mathcal{A} \times \mathcal{S} \to \mathbb{R}$ and continuation function

$\gamma : \mathcal{S} \times \mathcal{A} \times \mathcal{S} \to [0,1]$, with $C_{t+1} \overset{\text{def}}{=} c(S_t, A_t, S_{t+1})$ and $\gamma_{t+1} \overset{\text{def}}{=} \gamma(S_t, A_t, S_{t+1})$ for a (random) transition $(S_t, A_t, S_{t+1})$. The value for a state $s \in \mathcal{S}$ is

$$V(s) \overset{\text{def}}{=} \mathbb{E}_\pi [G_t | S_t = s] \qquad \text{where } G_t \overset{\text{def}}{=} C_{t+1} + \gamma_{t+1} C_{t+2} + \gamma_{t+1} \gamma_{t+2} C_{t+3} + \dots.$$

The operator $\mathbb{E}_\pi$ indicates an expectation with actions selected according to policy $\pi$. GVFs encompass standard value functions, where the cumulant is a reward. Otherwise, GVFs enable predictions about discounted sums of others signals into the future, when following a target policy $\pi$. These values are typically estimated using parametric function approximation, with weights $\theta \in \mathbb{R}^d$ defining approximate values $V_\theta(s)$.

In off-policy learning, transitions are sampled according to behavior policy, rather than the target policy. To get an unbiased sample of an update to the weights, the action probabilities need to be adjusted. Consider on-policy temporal difference (TD) learning, with update $\alpha_t \delta_t \nabla_\theta V_\theta(s)$ for a given $S_t = s$, for learning rate $\alpha_t \in \mathbb{R}^+$ and TD-error $\delta_t \overset{\text{def}}{=} C_{t+1} + \gamma_{t+1} V_\theta(S_{t+1}) - V_\theta(s)$. If actions are instead sampled according to a behavior policy $\mu : \mathcal{S} \times \mathcal{A} \to [0,1]$, then we can use importance sampling (IS) to modify the update, giving the off-policy TD update $\alpha_t \rho_t \delta_t \nabla_\theta V_\theta(s)$ for IS ratio $\rho_t \overset{\text{def}}{=} \frac{\pi(A_t|S_t)}{\mu(A_t|S_t)}$. Given state $S_t = s$, if $\mu(a|s) > 0$ when $\pi(a|s) > 0$, then the expected value of these two updates are equal. To see why, notice that

$$\mathbb{E}_\mu [\alpha_t \rho_t \delta_t \nabla_\theta V_\theta(s) | S_t = s] = \alpha_t \nabla_\theta V_\theta(s) \mathbb{E}_\mu [\rho_t \delta_t | S_t = s]$$

which equals $\mathbb{E}_\pi [\alpha_t \rho_t \delta_t \nabla_\theta V_\theta(s) | S_t = s]$ because

$$\mathbb{E}_\mu [\rho_t \delta_t | S_t = s] = \sum_{a \in \mathcal{A}} \mu(a|s) \frac{\pi(a|s)}{\mu(a|s)} \mathbb{E} [\delta_t | S_t = s, A_t = a] = \mathbb{E}_\pi [\delta_t | S_t = s].$$

Though unbiased, IS can be high-variance. A lower variance alternative is Weighted IS (WIS). For a batch consisting of transitions $\{(s_i, a_i, s_{i+1}, c_{i+1}, \rho_i)\}_{i=1}^n$, batch WIS uses a normalized estimate for the update. For example, an offline batch WIS TD algorithm, denoted WIS-Optimal below, would use update $\alpha_t \frac{\rho_t}{\sum_{i=1}^n \rho_i} \delta_t \nabla_\theta V_\theta(s)$. Obtaining an efficient WIS update is not straightforward, however, when learning online and has resulted in algorithms in the SGD setting (i.e. $n = 1$) specialized to tabular [Precup et al., 2001] and linear functions [Mahmood et al., 2014, Mahmood and Sutton, 2015]. We nonetheless use WIS as a baseline in the experiments and theory.

## 3 Importance Resampling

In this section, we introduce Importance Resampling (IR) for off-policy prediction and characterize its bias and variance. A resampling strategy requires a buffer of samples, from which we can resample. Replaying experience from a buffer was introduced as a biologically plausible way to reuse old experience [Lin, 1992, 1993], and has become common for improving sample efficiency, particularly for control [Mnih et al., 2015, Schaul et al., 2015b]. In the simplest case—which we assume here—the buffer is a sliding window of the most recent $n$ samples, $\{(s_i, a_i, s_{i+1}, c_{i+1}, \rho_i)\}_{i=t-n}^t$, at time step $t > n$. We assume samples are generated by taking actions according to behavior $\mu$. The transitions are generated with probability $d_\mu(s)\mu(a|s)\mathrm{P}(s'|s,a)$, where $d_\mu : \mathcal{S} \to [0,1]$ is the stationary distribution for policy $\mu$. The goal is to obtain samples according to $d_\mu(s)\pi(a|s)\mathrm{P}(s'|s,a)$, as if we had taken actions according to policy $\pi$ from states[3] $s \sim d_\mu$.

The IR algorithm is simple: resample a mini-batch of size $k$ on each step $t$ from the buffer of size $n$, proportionally to $\rho_i$ in the buffer. Using the resampled mini-batch we can update our value function using standard on-policy approaches, such as on-policy TD or on-policy gradient TD. The key difference to IS and WIS is that the distribution itself is corrected, before the update, whereas IS and WIS correct the update itself. This small difference, however, can have larger ramifications practically, as we show in this paper.

We consider two variants of IR: with and without bias correction. For point $i_j$ sampled from the buffer, let $\Delta_{i_j}$ be the on-policy update for that transition. For example, for TD, $\Delta_{i_j} = \delta_{i_j} \nabla_\theta V_\theta(s_{i_j})$. The first step for either variant is to sample a mini-batch of size $k$ from the buffer, proportionally to $\rho_i$. Bias-Corrected IR (BC-IR) additionally pre-multiplies with the average ratio in the buffer $\bar{\rho} \stackrel{\text{def}}{=} \frac{1}{n} \sum_{i=1}^n \rho_i$, giving the following estimators for the update direction

$$X_{\text{IR}} \stackrel{\text{def}}{=} \frac{1}{k} \sum_{j=1}^k \Delta_{i_j} \qquad\qquad X_{\text{BC}} \stackrel{\text{def}}{=} \frac{\bar{\rho}}{k} \sum_{j=1}^k \Delta_{i_j}$$

BC-IR negates bias introduced by the average ratio in the buffer deviating significantly from the true mean. For reasonably large buffers, $\bar{\rho}$ will be close to 1 making IR and BC-IR have near-identical updates[4]. Nonetheless, they do have different theoretical properties, particularly for small buffer sizes $n$, so we characterize both.

Across most results, we make the following assumption.

**Assumption 1.** *A buffer $B_t = \{X_{t+1}, ..., X_{t+n}\}$ is constructed from the most recent $n$ transitions sampled by time $t+n$, which are generated sequentially from an irreducible, finite MDP with a fixed policy $\mu$.*

To denote expectations under $p(x) = d_\mu(s)\mu(a|s)\text{P}(s'|s,a)$ and $q(x) = d_\mu(s)\pi(a|s)\text{P}(s'|s,a)$, we overload the notation from above, using operators $\mathbb{E}_\mu$ and $\mathbb{E}_\pi$ respectively. To reduce clutter, we write $\mathbb{E}$ to mean $\mathbb{E}_\mu$, because most expectations are under the sampling distribution. All proofs can be found in Appendix B.

## 3.1 Bias of IR

We first show that IR is biased, and that its bias is actually equal to WIS-Optimal, in Theorem 3.1.

**Theorem 3.1.** *[Bias for a fixed buffer of size $n$] Assume a buffer $B$ of $n$ transitions sampled i.i.d according to $p(x = (s, a, s')) = d_\mu(s)\mu(a|s)\text{P}(s'|s,a)$. Let $X_{\text{WIS}^*} \stackrel{\text{def}}{=} \sum_{i=1}^n \frac{\rho_i}{\sum_{j=1}^n \rho_j} \Delta_i$ be the WIS-Optimal estimator of the update. Then,*

$$\mathbb{E}[X_{\text{IR}}] = \mathbb{E}[X_{\text{WIS}^*}]$$

*and so the bias of $X_{\text{IR}}$ is proportional to*

$$\text{Bias}(X_{\text{IR}}) = \mathbb{E}[X_{\text{IR}}] - \mathbb{E}_\pi[\Delta] \propto \frac{1}{n}(\mathbb{E}_\pi[\Delta]\sigma_\rho^2 - \sigma_{\rho,\Delta}\sigma_\rho\sigma_\Delta) \tag{1}$$

*where $\mathbb{E}_\pi[\Delta]$ is the expected update across all transitions, with actions from $S$ taken by the target policy $\pi$; $\sigma_\rho^2 = \text{Var}(\frac{1}{n}\sum_{j=1}^n \rho_j)$; $\sigma_\Delta^2 = \text{Var}(\frac{1}{n}\sum_{i=1}^n \rho_i\Delta_i)$; and covariance $\sigma_{(\rho,\Delta)} = \text{Cov}(\frac{1}{n}\sum_{j=1}^n \rho_j, \frac{1}{n}\sum_{i=1}^n \rho_i\Delta_i)$.*

Theorem 3.1 is the only result which follows a different set of assumptions, primarily due to using the bias characterization of $X_{\text{WIS}^*}$ found in Owen [2013]. The bias of IR will be small for reasonably large $n$, both because it is proportional to $1/n$ and because larger $n$ will result in lower variance of the average ratios and average update for the buffer in Equation (1). In particular, as $n$ grows, these variances decay proportionally to $n$. Nonetheless, for smaller buffers, such bias could have an impact. We can, however, easily mitigate this bias with a bias-correction term, as shown in the next corollary and proven in Appendix B.2.

**Corollary 3.1.1.** *BC-IR is unbiased: $\mathbb{E}[X_{\text{BC}}] = \mathbb{E}_\pi[\Delta]$.*

## 3.2 Consistency of IR

Consistency of IR in terms of an increasing buffer, with $n \to \infty$, is a relatively straightforward extension of prior results for SIR, with or without the bias correction, and from the derived bias of both estimators (see Theorem B.1 in Appendix B.3). More interesting, and reflective of practice, is consistency *with a fixed length buffer* and increasing interactions with the environment, $t \to \infty$. IR, without bias correction, is asymptotically biased in this case; in fact, its asymptotic bias is the one characterized above for a fixed length buffer in Theorem 3.1. BC-IR, on the other hand, is consistent, even with a sliding window, as we show in the following theorem.

**Theorem 3.2.** *Let $B_t = \{X_{t+1}, ..., X_{t+n}\}$ be the buffer of the most recent $n$ transitions sampled according to Assumption 1. Define the sliding-window estimator $X_t \stackrel{\text{def}}{=} \frac{1}{T} \sum_{t=1}^{T} X_{\text{BC}}^{(t)}$. Then, if $\mathbb{E}_\pi[|\Delta|] < \infty$, then $X_T$ converges to $\mathbb{E}_\pi[\Delta]$ almost surely as $T \to \infty$.*

### 3.3 Variance of Updates

It might seem that resampling avoids high-variance in updates, because it does not reweight with large magnitude IS ratios. The notion of *effective sample size* from statistics, however, provides some intuition about why large magnitude IS ratios can also negatively affect IR, not just IS. Effective sample size is between 1 and $n$, with one estimator $\left(\sum_{i=1}^{n} \rho_i\right)^2 / \sum_{i=1}^{n} \rho_i^2$ [Kong et al., 1994, Martino et al., 2017]. When the effective sample size is low, this indicates that most of the probability is concentrated on a few samples. For high magnitude ratios, IR will repeatedly sample the same transitions, and potentially never sample some of the transitions with small IS ratios.

Fortunately, we find that, despite this dependence on effective sample size, IR can significantly reduce variance over IS. In this section, we characterize the variance of the BC-IR estimator. We choose this variant of IR, because it is unbiased and so characterizing its variance is a more fair comparison to IS. We define the mini-batch IS estimator $X_{\text{IS}} \stackrel{\text{def}}{=} \frac{1}{k} \sum_{j=1}^{k} \rho_{z_j} \Delta_{z_j}$, where indices $z_j$ are sampled uniformly from $\{1, \ldots, n\}$. This contrasts the indices $i_1, \ldots, i_k$ for $X_{\text{BC}}$ that are sampled proportionally to $\rho_i$.

We begin by characterizing the variance, under a fixed dataset $B$. For convenience, let $\mu_B = \mathbb{E}_\pi[\Delta|B]$. We characterize the sum of the variances of each component in the update estimator, which equivalently corresponds to normed deviation of the update from its mean,

$$\mathbb{V}(\Delta \mid B) \stackrel{\text{def}}{=} \text{tr Cov}(\Delta \mid B) = \sum_{m=1}^{d} \text{Var}(\Delta_m \mid B) = \mathbb{E}[\|\Delta - \mu_B\|_2^2 \mid B]$$

for an unbiased stochastic update $\Delta \in \mathbb{R}^d$. We show two theorems that BC-IR has lower variance than IS, with two different conditions on the norm of the update. We first start with more general conditions, and then provide a theorem for conditions that are likely only true in early learning.

**Theorem 3.3.** *Assume that, for a given buffer $B$, $\|\Delta_j\|_2^2 > \frac{c}{\rho_j}$ for samples where $\rho_j \geq \bar{\rho}$, and that $\|\Delta_j\|_2^2 < \frac{c}{\rho_j}$ for samples where $\rho_j < \bar{\rho}$, for some $c > 0$. Then the BC-IR estimator has lower variance than the IS estimator: $\mathbb{V}(X_{\text{BC}} \mid B) < \mathbb{V}(X_{\text{IS}} \mid B)$.*

The conditions in Theorem 3.3 preclude having update norms for samples with small $\rho$ be quite large—larger than a number $\propto \frac{1}{\rho}$—and a small norm for samples with large $\rho$. These conditions can be relaxed to a statement on average, where the cumulative weighted magnitude of the update norm for samples with $\rho$ below the median needs to be smaller than for samples with $\rho$ above the mean (see the proof in Appendix B.5).

We next consider a setting where the magnitude of the update is independent of the given state and action. We expect this condition to hold in early learning, where the weights are randomly initialized, and thus randomly incorrect across the state-action space. As learning progresses, and value estimates become more accurate in some states, it is unlikely for this condition to hold.

**Theorem 3.4.** *Assume $\rho$ and the magnitude of the update $\|\Delta\|_2^2$ are independent*

$$\mathbb{E}[\rho_j \|\Delta_j\|_2^2 \mid B] = \mathbb{E}[\rho_j \mid B] \, \mathbb{E}[\|\Delta_j\|_2^2 \mid B]$$

*Then the BC-IR estimator will have equal or lower variance than the IS estimator: $\mathbb{V}(X_{\text{BC}} \mid B) \leq \mathbb{V}(X_{\text{IS}} \mid B)$.*

These results have focused on variance of each estimator, for a fixed buffer, which provided insight into variance of updates when executing the algorithms. We would, however, also like to characterize variability across buffers, especially for smaller buffers. Fortunately, such a characterization is a simple extension on the above results, because variability for a given buffer already demonstrates variability due to different samples. It is easy to check that $\mathbb{E}[\mathbb{E}[\mu_{IR} \mid B]] = \mathbb{E}[\mu_{IS} \mid B] = \mathbb{E}_\pi[\Delta]$. The variances can be written using the law of total variance

$$\mathbb{V}(X_{\text{BC}}) = \mathbb{E}[\mathbb{V}(X_{\text{BC}} \mid B)] + \mathbb{V}(\mathbb{E}[X_{\text{BC}} \mid B]) = \mathbb{E}[\mathbb{V}(X_{\text{BC}} \mid B)] + \mathbb{V}(\mu_B)$$

$$\mathbb{V}(X_{\text{IS}}) = \mathbb{E}[\mathbb{V}(X_{\text{IS}} \mid B)] + \mathbb{V}(\mu_B)$$

$$\implies \mathbb{V}(X_{\text{BC}}) - \mathbb{V}(X_{\text{IS}}) = \mathbb{E}[\mathbb{V}(X_{\text{BC}} \mid B) - \mathbb{V}(X_{\text{IS}} \mid B)]$$

with expectation across buffers. Therefore, the analysis of $\mathbb{V}(X_{\text{BC}} \mid B)$ directly applies.

# 4 Empirical Results

We investigate the two hypothesized benefits of resampling as compared to reweighting: improved sample efficiency and reduced variance. These benefits are tested in two microworld domains— a Markov chain and the Four Rooms domain—where exhaustive experiments can be conducted. We also provide a demonstration that IR reduces sensitivity over IS and VTrace in a car simulator, TORCs, when learning from images [5].

We compare IR and BC-IR against several reweighting strategies, including importance sampling (IS); two online approaches to weighted important sampling, WIS-Minibatch with weighting $\rho_i / \sum_{j=1}^{k} \rho_j$ and WIS-Buffer with weighting $\rho_i / \frac{k}{n} \sum_{j=1}^{n} \rho_j$; and V-trace[6], which corresponds to clipping importance weights [Espeholt et al., 2018]. We also compare to WIS-TD(0) [Mahmood and Sutton, 2015], when applicable, which uses an online approximation to WIS, with a stepsize selection strategy (as described in Appendix A.2). This algorithm uses only one sample at a time, rather than a mini-batch, and so is only included in Figure 2. Where appropriate, we also include baselines using On-policy sampling; WIS-Optimal which uses the whole buffer to get an update; and Sarsa(0) which learns action-values—which does not require IS ratios—and then produces estimate $V(s) = \sum_a \pi(s,a)Q(s,a)$. WIS-Optimal is included as an optimal baseline, rather than as a competitor, as it estimates the update using the whole buffer on every step.

In all the experiments, the data is generated off-policy. We compute the absolute value error (AVE) or the absolute return error (ARE) on every step. For the sensitivity plots we take the average over all the interactions as specified for the environment — resulting in MAVE and MARE respectively. The error bars represent the standard error over runs, which are featured on every plot — although not visible in some instances. For the microworlds, the true value function is found using dynamic programming with threshold $10^{-15}$, and we compute AVE over all the states. For TORCs and continuous Four Rooms, the true value function is approximated using rollouts from a random subset of states generated when running the behavior policy $\mu$, and the ARE is computed over this subset. For the Torcs domain, the same subset of states is used for each run due to computational constraints and report the mean squared return error (MSRE). Plots showing sensitivity over number of updates show results for complete experiments with updates evenly spread over all the interactions. A tabular representation is used in the microworld experiments, tilecoded features with 64 tilings and 8 tiles is used in continuous Four Rooms, and a convolutional neural network is used for TORCs, with an architecture previously defined for self-driving cars [Bojarski et al., 2016].

## 4.1 Investigating Convergence Rate

We first investigate the convergence rate of IR. We report learning curves in Four Rooms, as well as sensitivity to the learning rate. The Four Rooms domain [Stolle and Precup, 2002] has four rooms in an 11x11 grid world. The four rooms are positioned in a grid pattern with each room having two adjacent rooms. Each adjacent room is separated by a wall with a single connecting hallway. The target policy takes the down action deterministically. The cumulant for the value function is 1 when the agent hits a wall and 0 otherwise. The continuation function is $\gamma = 0.9$, with termination when the agent hits a wall. The resulting value function can be thought of as distance to the bottom wall. The behavior policy is uniform random everywhere except for 25 randomly selected states which take the action down with probability 0.05 with remaining probability split equally amongst the other actions. The choice of behavior and target policy induce high magnitude IS ratios.

As shown in Figure 1, IR has noticeable improvements over the reweighting strategies tested. The fact that IR resamples more important transitions from the replay buffer seems to significantly increase the learning speed. Further, IR has a wider range of usable learning rates. The same effect is seen even as we reduce the total number of updates, where the uniform sampling methods perform significantly worse as the interactions between updates increases—suggesting improved sample efficiency. WIS-Buffer performs almost equivalently to IS, because for reasonably size buffers, its normalization factor $\frac{1}{n} \sum_{j=1}^{n} \rho_j \approx 1$ because $\mathbb{E}[\rho] = 1$. WIS-Minibatch and V-trace both reduce

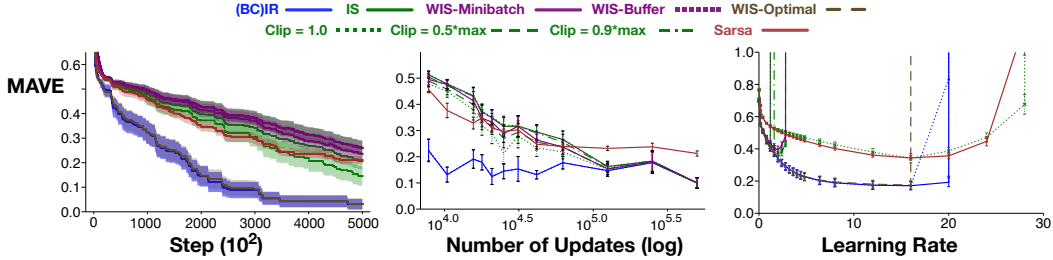

Figure 1: Four Rooms experiments ($n = 2500$, $k = 16$, 25 runs): **left** Learning curves for each method, with updates every 16 steps. IR and WIS-Optimal are overlapping. **center** Sensitivity over the number of interactions between updates. **right** Learning rate sensitivity plot.

the variance significantly, with their bias having only a limited impact on the final performance compared to IS. Even the most aggressive clipping parameter for V-trace—a clipping of 1.0— outperforms IS. The bias may have limited impact because the target policy is deterministic, and so only updates for exactly one action in a state. Sarsa—which is the same as Retrace(0)—performs similarly to the reweighting strategies.

The above results highlight the convergence rate improvements from IR, in terms of number of updates, without generalization across values. Conclusions might actually be different with function approximation, when updates for one state can be informative for others. For example, even if in one state the target policy differs significantly from the behavior policy, if they are similar in a related state, generalization could overcome effective sample size issues. We therefore further investigate if the above phenomena arise under function approximation with RMSProp learning rate selection.

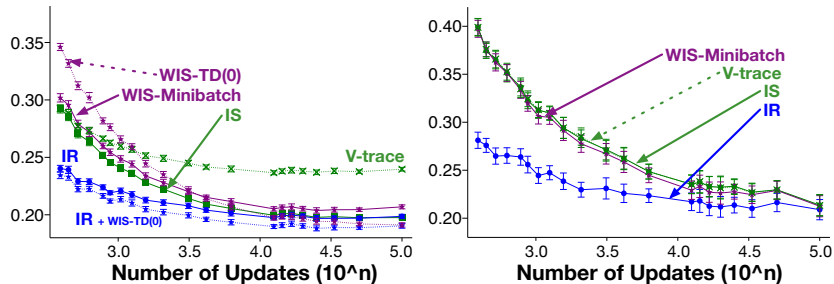

Figure 2: Convergence rates in Continuous Four Rooms averaged over 25 runs with 100000 interactions with the environment. **left** uniform random behavior policy and target policy which takes the down action with probability 0.9 and probability $0.1/3$ for all other actions. Learning used incremental updates (as specified in appendix A.2). **right** uniform random behavior and target policy with persistent down action selection learned with mini-batch updates with RMSProp.

We conduct two experiments similar to above, in a continuous state Four Rooms variant. The agent is a circle with radius 0.1, and the state consists of a continuous tuple containing the x and y coordinates of the agent's center point. The agent takes an action in one of the 4 cardinal directions moving $0.5 \pm \mathcal{U}(0.0, 0.1)$ in that directions with random drift in the orthogonal direction sampled from $\mathcal{N}(0.0, 0.01)$. The representation is a tile coded feature vector with 64 tilings and 8 tiles. We provide results for both mini-batch updating (as above) and incremental updating (i.e. updating on each transition of a mini-batch incrementally, see appendix A.2 for details). For the mini-batch experiment, the target policy deterministically takes the down action. For the incremental experiment, the target policy takes the down action with probability 0.9 and selects all other action with probability $0.1/3$.

We find that generalization can mitigate some of the differences between IR and IS above in some settings, but in others the difference remains just as stark (see Figure 2 and Appendix C.2). If we use the behavior policy from the tabular domain, which skews the behavior in a sparse set of states, the nearby states mitigate this skew. However, if we use a behavior policy that selects all actions

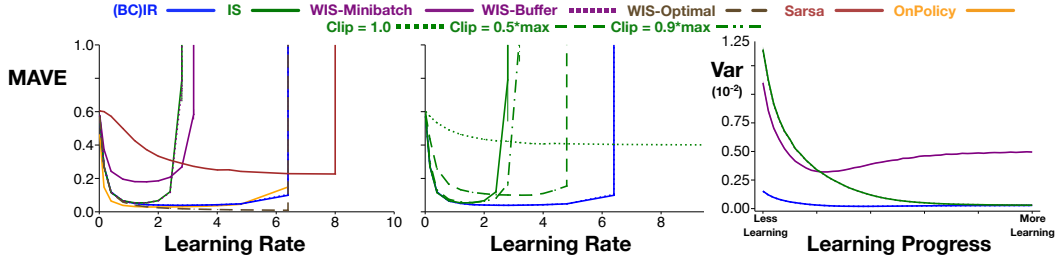

Figure 4: Learning Rate sensitivity plots in the Random Walk Markov Chain, with buffer size $n = 15000$ and mini-batch size $k = 16$. Averaged over 100 runs. The policies, written as [probability left, probability right] are $\mu = [0.9, 0.1], \pi = [0.1, 0.9]$ **left** learning rate sensitivity plot for all methods but V-trace. **center** learning rate sensitivity for V-trace with various clipping parameters **right** Variance study for IS, IR, and WISBatch. The x-axis corresponds to the training iteration, with variance reported for the weights at that iteration generated by WIS-Optimal. These plots show a correlation between the sensitivity to learning rate and magnitude of variance.

uniformly, then again IR obtains noticeable gains over IS and V-trace, for reducing the required number of updates, as shown in Figure 2.

We find similar results for the incremental setting Figure 2 (left), where resampling still outperforms all other methods in terms of convergence rates. Given WIS-TD(0)'s significant degrade in performance as the number of updates decreases, we also compare with using WIS-TD(0) when sampling according to resampling IR+WIS-TD(0). Interestingly, this method outperforms all others — albeit only slightly against IR with constant learning rate. This result leads us to believe RMSProp may be a optimizer poor choice for this setting. Expanded results can be found in Appendix C.2.

## 4.2 Investigating Variance

To better investigate the update variance we use a Markov chain, where we can more easily control dissimilarity between $\mu$ and $\pi$, and so control the magnitude of the IS ratios. The Markov chain is composed of 8 non-terminating states and 2 terminating states on the ends of the chain, with a cumulant of 1 on the transition to the right-most terminal state and 0 everywhere else. We consider policies with probabilities [left, right] equal in all states: $\mu = [0.9, 0.1], \pi = [0.1, 0.9]$; further policy settings can be found in Appendix C.1.

We first measure the variance of the updates for fixed buffers. We compute the variance of the update—from a given weight vector—by simulating the many possible updates that could have occurred. We are interested in the variance of updates both for early learning—when the weight vector is quite incorrect and updates are larger—and later learning. To obtain a sequence of such weight vectors, we use the sequence of weights generated by WIS-Optimal. As shown in Figure 4, the variance of IR is lower than IS, particularly in early learning, where the difference is stark. Once the weight vector has largely converged, the variance of IR and IS is comparable and near zero.

We can also evaluate the update variance by proxy using learning rate sensitivity curves. As seen in Figure 4 (left) and (center), IR has the lowest sensitivity to learning rates, on-par with On-Policy sampling. IS has the highest sensitivity, along with WIS-Buffer and WIS-Minibatch. Various clipping parameters with V-trace are also tested. V-trace does provide some level of variance reduction but incurs more bias as the clipping becomes more aggressive.

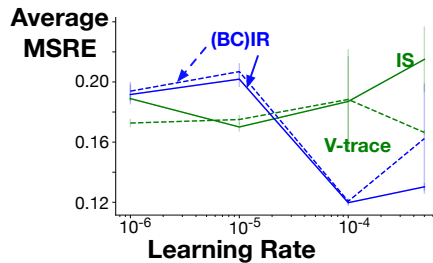

Figure 3: Learning rate sensitivity in TORCs, averaged over 10 runs. V-trace has clipping parameter 1.0. All the methods performed worse with a higher learning rate than shown here, so we restrict to this range.

## 4.3 Demonstration on a Car Simulator

We use the TORCs racing car simulator to perform scaling experiments with neural networks to compare IR, IS, and V-trace. The simulator produces

64x128 cropped grayscale images. We use an underlying deterministic steering controller that produces steering actions $a_{det} \in [-1, +1]$ and take an action with probability defined by a Gaussian $a \sim \mathcal{N}(a_{det}, 0.1)$. The target policy is a Gaussian $\mathcal{N}(0.15, 0.0075)$, which corresponds to steering left. Pseudo-termination (i.e., $\gamma = 0$) occurs when the car nears the center of the road, and the cumulant becomes 1. Otherwise, the cumulant is zero and $\gamma = 0.9$. The policy is specified using continuous action distributions and results in IS ratios as high as $\sim 1000$ and highly variant updates for IS.

Again, we can see that IR provides benefits over IS and V-trace, in Figure 3. There is even more generalization from the neural network in this domain, than in Four Rooms where we found generalization did reduce some of the differences between IR and IS. Yet, IR still obtains the best performance, and avoids some of the variance seen in IS for two of the learning rates. Additionally, BC-IR actually performs differently here, having worse performance for the largest learning rate. This suggest IR has an effect in reducing variance.

## 5   Conclusion

In this paper we introduced a new approach to off-policy learning: Importance Resampling. We showed that IR is consistent, and that the bias is the same as for Weighted Importance Sampling. We also provided an unbiased variant of IR, called Bias-Corrected IR. We empirically showed that IR (a) has lower learning rate sensitivity than IS and V-trace, which is IS with varying clipping thresholds; (b) the variance of updates for IR are much lower in early learning than IS and (c) IR converges faster than IS and other competitors, in terms of the number of updates. These results confirm the theory presented for IR, which states that variance of updates for IR are lower than IS in two settings, one being an early learning setting. Such lower variance also explains why IR can converge faster in terms of number of updates, for a given buffer of data.

The algorithm and results in this paper suggest new directions for off-policy prediction, particularly for faster convergence. Resampling is promising for scaling to learning many value functions in parallel, because many fewer updates can be made for each value function. A natural next step is a demonstration of IR, in such a parallel prediction system. Resampling from a buffer also opens up questions about how to further focus updates. One such option is using an intermediate sampling policy. Another option is including prioritization based on error, such as was done for control with prioritized sweeping [Peng and Williams, 1993] and prioritized replay [Schaul et al., 2015b].

**Acknowledgments**

We would like to thank Huawei for their support, and especially for allowing a portion of this work to be completed during Matthews internship in the summer of 2018. We also would like to acknowledge University of Alberta, Alberta Machine Intelligence Institute, IVADO, and NSERC for their continued funding and support, as well as Compute Canada (`www.computecanada.ca`) for the computing resources used for this work.

## Footnotes

[1]There is substantial literature on variance reduction for another area called off-policy policy evaluation, but which estimates only a single number or value for a policy (e.g., see [Thomas and Brunskill, 2016]). The resulting algorithms differ substantially, and are not appropriate for learning the value function.

[2]We explicitly use the term prediction rather than policy evaluation to make it clear that we are not learning value functions for control. Rather, our goal is to learn value functions solely for the sake of prediction.

[3]The assumption that states are sampled from $d_\mu$ underlies most off-policy learning algorithms. Only a few attempt to adjust probabilities $d_\mu$ to $d_\pi$, either by multiplying IS ratios before a transition [Precup et al., 2001] or by directly estimating state distributions [Hallak and Mannor, 2017, Liu et al., 2018]. In this work, we focus on using resampling to correct the action distribution—the standard setting. We expect, however, that some insights will extend to how to use resampling to correct the state distribution, particularly because wherever IS ratios are used it should be straightforward to use our resampling approach.

[4] $\bar{\rho} \approx \mathbb{E}[\rho(a|s)] = \mathbb{E}[\frac{\pi(a|s)}{\mu(a|s)}] = \sum_{s,a} \frac{\pi(a|s)}{\mu(a|s)} \mu(a|s) d_\mu(s) = 1.$

[5]Experimental code for every domain except Torcs can be found at https://mkschleg.github.io/Resampling.jl

[6]Retrace, ABQ and TreeBackup also use clipping to reduce variance. But, they are designed for learning action-values and for mitigating variance in eligibility traces. When trace parameter $\lambda = 0$—as we assume here—there are no IS ratios and these methods become equivalent to using Sarsa(0) for learning action-values.

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
