[Supplementary Material · paper_final_camera_ready.pdf]

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

## A Weighted Importance Sampling

### A.1 Mini-Batch Algorithms

We consider three weighted importance sampling updates as competitors to IR. $n$ is the size of the experience replay buffer, $k$ is the size of a single batch. WIS-Minibatch and WIS-Buffer both follow a similar protocol as IS, in that they uniformly sample a mini-batch from the experience replay buffer and use this to update the value functions. The difference comes in the scaling of the update. The first, WIS-Minibatch, uses the sum of the importance weights $\rho_i$ in the sampled mini-batch, while WIS-Buffer uses the sum of importance weights in the experience replay buffer. WIS-Buffer is also scaled by the size of the buffer and brought to the same effective scale as the other updates with $\frac{1}{k}$. WIS-Optimal follows a different approach and performs the best possible version of WIS where the gradient descent update is calculated from the whole experience replay buffer. We do not provide analysis on the bias or consistency of WIS-Minibatch or WIS-Buffer, but are natural versions of WIS one might try.

$$\Delta\theta = \frac{\sum_i^k \rho_i \delta_i \nabla_\theta V(s_i; \theta)}{\sum_j^k \rho_j} \qquad \text{WIS-Minibatch}$$

$$\Delta\theta = \frac{n}{k} \frac{\sum_i^k \rho_i \delta_i \nabla_\theta V(s_i; \theta)}{\sum_j^n \rho_j} \qquad \text{WIS-Buffer}$$

$$\Delta\theta = \frac{\sum_i^n \rho_i \delta_i \nabla_\theta V(s_i; \theta)}{\sum_j^n \rho_j} \qquad \text{WIS-Optimal}$$

### A.2 Incremental Algorithm

While implementing an efficient true WIS algorithm for mini-batch updating is beyond the scope of this work, we compare WIS-TD(0) to the incremental versions of IR, IS, VTrace, and WISBatch. The difference between the mini-batch and incremental algorithms is how the updates are calculated. In the incremental scheme a random mini-batch of data is similarly sampled from the buffer. We then use each sample individually to update our value function. We do this to more naturally compare our baselines to WIS-TD(0) Mahmood and Sutton [2015]. WIS-TD(0) has parameters $u_0 \in \{\frac{1}{64}, 1, 5, 10, 50\} * 64, \mu \in 10^{-2:0.25:1}$, and $\eta = \frac{\mu}{u_0}$. WIS-TD(0) follows the update equations:

$$\mathbf{u}_{i+1} = (\mathbf{1} - \eta\phi_i \circ \phi_i) \circ \mathbf{u}_i + \rho_i \phi_i \circ \phi_i \quad \triangleright \circ \overset{\text{def}}{=} \text{ element wise product}$$

$$\alpha_{i+1} = \mathbf{1} \oslash \mathbf{u}_{t+1} \quad \triangleright \oslash \overset{\text{def}}{=} \text{ element wise division}$$

$$\bar{\delta}_i = C_i + \gamma_i \boldsymbol{\theta}_i^\top \phi_i' - \boldsymbol{\theta}_{i-1}^\top \phi_i$$

$$\boldsymbol{\theta}_{i+1} = \boldsymbol{\theta}_t + \boldsymbol{\alpha}_{i+1} \circ \rho_i (\boldsymbol{\theta}_{i-1}^\top \phi_i - \boldsymbol{\theta}_i^\top \phi_i)\phi_i + \rho_i \bar{\delta}_i \boldsymbol{\alpha}_{i+1} \circ \phi_i$$

where $\boldsymbol{\theta} \in \mathbb{R}^d$ is the weight vector of the value function, $\phi_i \in \mathbb{R}^d$ is the feature vector of the i-th transition in the experience replay buffer, and $\phi_i' \in \mathbb{R}^d$ is the feature vector of the next state of the i-th transition in the experience replay buffer.

## B Additional Theoretical Results and Proofs

### B.1 Bias of IR

**Theorem 3.1**(Bias for a fixed buffer of size $n$) Assume a buffer $B$ of $n$ transitions sampled i.i.d according to $p(x = (s, a, s')) = d_\mu(s)\mu(a|s)\mathrm{P}(s'|s, a)$. Let $X_{\mathrm{WIS}^*} \overset{\text{def}}{=} \sum_{i=1}^n \frac{\rho_i}{\sum_{j=1}^n \rho_j}\Delta_i$ be the WIS-Optimal estimator of the update. Then,

$$\mathbb{E}[X_{\mathrm{IR}}] = \mathbb{E}[X_{\mathrm{WIS}^*}]$$

and so the bias of $X_{\mathrm{IR}}$ is proportional to

$$\text{Bias}(X_{\text{IR}}) = \mathbb{E}[X_{\text{IR}}] - \mathbb{E}_\pi[\Delta] \propto \frac{1}{n}(\mathbb{E}_\pi[\Delta]\sigma_\rho^2 - \sigma_{\rho,\Delta}\sigma_\rho\sigma_\Delta)$$

where $\mathbb{E}_\pi[\Delta]$ is the expected update across all transitions, with actions from $S$ taken by the target policy $\pi$; $\sigma_\rho^2 = \text{Var}(\frac{1}{n}\sum_{j=1}^n \rho_j)$; $\sigma_\Delta^2 = \text{Var}(\frac{1}{n}\sum_{i=1}^n \rho_i\Delta_i)$; and covariance $\sigma_{(\rho,\Delta)} = \text{Cov}(\frac{1}{n}\sum_{j=1}^n \rho_j, \frac{1}{n}\sum_{i=1}^n \rho_i\Delta_i)$.

*Proof.* Notice first that when we weight with $\rho_i$, this is equivalent to weighting with $\frac{d_\mu(S_i)\pi(A_i|S_i)\text{P}(S_{i+1}|S_i,A_i)}{d_\mu(S_i)\mu(A_i|S_i)\text{P}(S_{i+1}|S_i,A_i)}$, and so is the correct IS ratio for the transition.

$$\mathbb{E}[X_{\text{IR}}] = \mathbb{E}\left[\mathbb{E}[X_{\text{IR}}|B]\right] = \mathbb{E}\left[\mathbb{E}\left[\frac{1}{k}\sum_{j=1}^k \Delta_{i_j}|B\right]\right]$$

$$= \mathbb{E}\left[\frac{1}{k}\sum_{j=1}^k \mathbb{E}[\Delta_{i_j}|B]\right] \qquad \triangleright \mathbb{E}[\Delta_{i_j}|B] = \sum_{i=1}^n \frac{\rho_i}{\sum_{j=1}^n \rho_j}\Delta_i$$

$$= \mathbb{E}\left[\sum_{i=1}^n \frac{\rho_i}{\sum_{j=1}^n \rho_j}\Delta_i\right]$$

$$= \mathbb{E}[X_{\text{WIS}^*}].$$

This bias of $X_{\text{IR}}$ is the same as $X_{\text{WIS}^*}$, which is characterized in Owen [2013], completing the proof. $\qquad\square$

## B.2 Proof of Unbiasedness of BC-IR

**Corollary 3.1.1** BC-IR is unbiased: $\mathbb{E}[X_{\text{BC}}] = \mathbb{E}_\pi[\Delta]$.

*Proof.*

$$\mathbb{E}[X_{\text{BC}}] = \mathbb{E}\left[\frac{\bar{\rho}}{k}\sum_{j=1}^k \mathbb{E}[\Delta_{i_j}|B]\right] = \mathbb{E}\left[\bar{\rho}\sum_{i=1}^n \frac{\rho_i}{\sum_{j=1}^n \rho_j}\Delta_i\right]$$

$$= \mathbb{E}\left[\frac{1}{n}\sum_{i=1}^n \rho_i\Delta_i\right] = \frac{1}{n}\sum_{i=1}^n \mathbb{E}\left[\frac{\pi(A_i|S_i)}{\mu(A_i|S_i)}\Delta_i\right]$$

$$= \frac{1}{n}\sum_{i=1}^n \mathbb{E}\left[\frac{d_\mu(S_i)\pi(A_i|S_i)\text{P}(S_{i+1}|S_i,A_i)}{d_\mu(S_i)\mu(A_i|S_i)\text{P}(S_{i+1}|S_i,A_i)}\Delta_i\right]$$

$$= \frac{1}{n}\sum_{i=1}^n \mathbb{E}_\pi[\Delta_i] = \mathbb{E}_\pi[\Delta].$$

The last equality follows from the fact that the samples are identically distributed. $\qquad\square$

## B.3 Consistency of the resampling distribution with a growing buffer

We show that the distribution when following a resampling strategy is consistent: as $n \to \infty$, the resampling distribution converges to the true distribution. Our approach closely follows that of [Smith et al., 1992], but we recreate it here for convenience.

**Proposition B.1.** *Let $X_n = \{x_1, x_2, ..., x_n\}$ be a buffer of data sampled i.i.d. according to proposal density $p(x_i)$. Let $q(x_i)$ be some distribution of interest with associated random variable $Q$ and assume the proposal distribution samples everywhere where $q(\cdot)$ is non-zero, i.e $supp(q) \subseteq supp(p)$. Also, let $Y$ be a discrete random variable taking values $x_i$ with probability $\propto \frac{q(x_i)}{p(x_i)}$. Then, $Y$ converges in distribution to $Q$ as $n \to \infty$.*

*Proof.* Let $\rho_i = \frac{q(x_i)}{p(x_i)}$. From the probability mass function of $Y$, we have that:

$$\mathbb{P}[Y \leq a] = \sum_{i=1}^{n} \mathbb{P}[Y = x_i] \mathbb{1}\{x_i \leq a\}$$

$$= \frac{n^{-1} \sum_{i=1}^{n} \rho_i \mathbb{1}\{x_i \leq a\}}{n^{-1} \sum_{i=1}^{n} \rho_i}$$

$$\xrightarrow{n \to \infty} \frac{\mathbb{E}_q[\rho(x) \mathbb{1}\{x \leq a\}]}{\mathbb{E}_q[\rho(x)]}$$

$$= \frac{1 \cdot \int_{-\infty}^{a} \frac{q(x)}{p(x)} p(x) dx + 0 \cdot \int_{a}^{\infty} \frac{q(x)}{p(x)} p(x) dx}{\int_{-\infty}^{\infty} \frac{q(x)}{p(x)} p(x) dx}$$

$$= \int_{-\infty}^{a} q(x) dx = \mathbb{P}[Q \leq a]$$

$$Y \xrightarrow{d} Q$$

$\square$

This means a resampling strategy effectively changes the distribution of random variable $X_n$ to that of $q(x)$, meaning we can use samples from $Y$ to build statistics about the target distribution $q(x)$. This result motivates using resampling to correct the action distribution in off-policy learning. This result can also be used to show that the IR estimators are consistent, with $n \to \infty$.

### B.4 Consistency under a sliding window

**Lemma B.2.** *Let $B_t = \{X_{t+1}, ..., X_{t+n}\}$ be the buffer of the most recent $n$ transitions sampled by time $t + n$, which are generated sequentially from an irreducible, finite MDP with a fixed policy $\mu$. We define $X_{\mathrm{BC}}^{(t)}$ be the BCIR estimator for buffer $B_t$. If $\mathbb{E}_\pi[|\Delta|] < \infty$, then $\mathbb{E}[X_{\mathrm{BC}}^{(t)}] = \mathbb{E}_\pi[\Delta]$.*

*Proof.* Let $X_t = (S_t, A_t, R_{t+1}, S_{t+1})$ be a transition and $\{B_t\}_{t \in \mathbb{N}}$ be the sequence of buffers that are observed, each containing $n$ consecutive transitions.

Using the law of iterated expectations,

$$\mathbb{E}\left[X_{\mathrm{BC}}^{(t)}\right] = \mathbb{E}\left[\mathbb{E}[X_{\mathrm{BC}}^{(t)} | B_t]\right]$$

where the outer expectation is over the stationary distribution of $B_t$ and the inner expectation is over the sampling distribution of IR from the buffer $B_t$.

Using the definition of $X_{\mathrm{BC}}^{(t)} | B_t$, we have that

$$\mathbb{E}[X_{\mathrm{BC}}^{(t)} | B_t] = \bar{\rho} \sum_{i=1}^{n} \Delta_i \frac{\rho_i}{\sum_{i=1}^{n} \rho_i}$$

$$= \frac{1}{n} \sum_{i=1}^{n} \rho_i \Delta_i$$

Next, the stationary distribution of $B_t$ is given by $d(B_t) = Pr(B_t = (x_{t+1}, ..., x_{t+n})) = d_X(x_t) p(x_{t+1} | x_t) ... p(x_{t+n} | x_{t+n-1})$, where $d_X$ is the stationary distribution of $X_t$. We can verify directly by checking that for all $B' = (x_2, ..., x_{n+1})$

$$\sum_{B} d(B) p(B' | B) = d(B')$$

where $B = (x_1, ..., x_n)$

To see this, first note that $p(B'|B) = p(x_{n+1}|x_n)\mathbf{1}(B, B')$ where $\mathbf{1}(B, B')$ is equal to 1 if the states $(x_2, ..., x_n)$ in $B$ match the states $(x_2, .., x_n)$ in $B'$. In other words, the first $n - 1$ transitions in $B'$

must match the last $n-1$ transitions in $B$ for $p(B'|B)$ to be positive. Next, fixing $B'$,

$$\sum_B d(B)p(B'|B)$$

$$= \sum_{x_1,\dots,x_n} d_\mu(x_1)p(x_2|x_1)\dots p(x_n|x_{n-1})p(x_{n+1}|x_n)\mathbf{1}(B,B')$$

$$= \sum_{x_1} d_\mu(x_1)p(x_2|x_1)\dots p(x_n|x_{n-1})p(x_{n+1}|x_n) \quad \text{since } (x_2,\dots,x_n) \text{ have to match}$$

$$= d_\mu(x_2)p(x_3|x_2)\dots p(x_n|x_{n-1})p(x_{n+1}|x_n)$$

$$= d(B')$$

which verifies the expression for the stationary distribution of $B_t$.

Continuing from before, we expand the expectation as:

$$\mathbb{E}\left[\frac{1}{n}\sum_{t=1}^{n}\rho_t\Delta_t\right]$$

$$= \sum_{x_1,\dots x_n} d_X(x_1)p(x_2|x_1)\dots p(x_n|x_{n-1})\left(\frac{1}{n}\sum_{t=1}^{n}\rho_t\Delta_t\right)$$

$$= \sum_{\substack{s_1,a_1,r_2,s_2,\dots,\\ s_n,a_n,r_{n+1},s_{n+1}}} d_\mu(s_1)\left(\prod_{i=1}^{n}\mu(a_i|s_i)p(s_{i+1},r_{i+1}|s_i,a_i)\right)\left(\frac{1}{n}\sum_{t=1}^{n}\rho_t\Delta_t\right)$$

$$= \frac{1}{n}\sum_{t=1}^{n}\sum_{\substack{s_1,a_1,r_2,s_2,\dots,\\ s_n,a_n,r_{n+1},s_{n+1}}} d_\mu(s_1)\left(\prod_{i=1}^{n}\mu(a_i|s_i)p(s_{i+1},r_{i+1}|s_i,a_i)\right)(\rho_t\Delta_t).$$

Next, by taking the sums over $(s_1,a_1,\dots r_{n+1},s_{n+1})$ within the products to make the summands depend only on the variables being summed over, we get

$$= \frac{1}{n}\sum_{t=1}^{n}\sum_{s_1} d_\mu(s_1)\sum_{a_1,r_2,s_2}\mu(a_1|s_1)p(s_2,r_2|s_1,a_1)\sum_{a_2,r_3,s_3}\mu(a_2|s_2)p(s_3,r_3|s_2,a_2)\dots$$

$$\sum_{a_t,r_{t+1},s_{t+1}}\mu(a_t|s_t)p(s_{t+1},r_{t+1}|s_t,a_t)(\rho_t\Delta_t)$$

$$\sum_{\substack{a_{t+1},r_{t+2},s_{t+2},\dots,\\ s_n,a_n,r_{n+1},s_{n+1}}}\prod_{i=t+1}^{n}\mu(a_i|s_i)p(s_{i+1},r_{i+1}|s_i,a_i)$$

$$= \frac{1}{n}\sum_{t=1}^{n}\sum_{s_1} d_\mu(s_1)\sum_{a_1,r_2,s_2}\mu(a_1|s_1)p(s_2,r_2|s_1,a_1)\sum_{a_2,r_3,s_3}\mu(a_2|s_2)p(s_3,r_3|s_2,a_2)\dots$$

$$\sum_{a_t,r_{t+1},s_{t+1}}\mu(a_t|s_t)p(s_{t+1},r_{t+1}|s_t,a_t)(\rho_t\Delta_t).$$

This followed since the third line is summing over the probability of all trajectories starting from $s_{t+1}$ and thus is equal to 1. Next, we note that, if $C$ is a constant that does not depend on $s_1, a_1, r_2$, then $\sum_{s_1,a_1,r_2} d_\mu(s_1)\mu(a_1|s_1)p(s_2,r_2|s_1,a_1)C = d_\mu(s_2)C$ since $d_\mu(s_2)$ is the stationary distribution (if we additionally assume $p(s_2,r_2|s_1,a_1) = p(s_2|s_1,a_1)p(r_2|s_1,a_1)$ or equivalently that rewards depend only on state and action).

Continuing from before, by reordering the sums we have and repeatedly using the above note,

$$
= \frac{1}{n} \sum_{t=1}^{n} \sum_{s_2} \underbrace{\sum_{s_1,a_1,r_2} d_\mu(s_1)\mu(a_1|s_1)p(s_2,r_2|s_1,a_1)}_{d_\mu(s_2)} \sum_{a_2,r_3,s_3} \mu(a_2|s_2)p(s_3,r_3|s_2,a_2)...
$$

$$
\sum_{a_t,r_{t+1},s_{t+1}} \mu(a_t|s_t)p(s_{t+1},r_{t+1}|s_t,a_t)\left(\rho_t\Delta_t\right)
$$

$$
= \frac{1}{n} \sum_{t=1}^{n} \sum_{s_2} d_\mu(s_2) \sum_{a_2,r_3,s_3} \mu(a_2|s_2)p(s_3,r_3|s_2,a_2)...
$$

$$
\sum_{a_t,r_{t+1},s_{t+1}} \mu(a_t|s_t)p(s_{t+1},r_{t+1}|s_t,a_t)\left(\rho_t\Delta_t\right)
$$

$$
= ... \quad \text{(Repeating the same process)}
$$

$$
= \frac{1}{n} \sum_{t=1}^{n} \sum_{s_t,a_t,r_{t+1},s_{t+1}} d_\mu(s_t)\mu(a_t|s_t)p(s_{t+1},r_{t+1}|s_t,a_t)\left(\rho_t\Delta_t\right).
$$

Recall that $\Delta_t = \Delta(s_t,a_t,r_{t+1},s_{t+1})$ is a function of the transition so we cannot simplify further.

Finally,

$$
= \frac{1}{n} \sum_{t=1}^{n} \sum_{s_t,a_t,r_{t+1},s_{t+1}} d_\mu(s_t)\mu(a_t|s_t)p(s_{t+1},r_{t+1}|s_t,a_t)\left(\frac{\pi(a_t)}{\mu(a_t)}\Delta_t\right)
$$

$$
= \frac{1}{n} \sum_{t=1}^{n} \sum_{s_t,a_t,r_{t+1},s_{t+1}} d_\mu(s_t)\pi(a_t|s_t)p(s_{t+1},r_{t+1}|s_t,a_t)\Delta_t
$$

$$
= \frac{1}{n} \sum_{t=1}^{n} \mathbb{E}_\pi[\Delta]
$$

$$
= \mathbb{E}_\pi[\Delta]
$$

$\square$

**Theorem 3.2** Let $B_t = \{X_{t+1},...,X_{t+n}\}$ be the buffer of the most recent $n$ transitions sampled by time $t+n$, which are generated sequentially from an irreducible, finite MDP with a fixed policy $\mu$. Define the sliding-window estimator $X_t \stackrel{\text{def}}{=} \frac{1}{T}\sum_{t=1}^{T} X_{\text{BC}}^{(t)}$. Then, if $\mathbb{E}_\pi[|\Delta|] < \infty$, then $X_T$ converges to $\mathbb{E}_\pi[\Delta]$ almost surely as $T \to \infty$.

*Proof.* Let $X_t = (S_t, A_t, R_{t+1}, S_{t+1})$ be a transition. Then the sequence $\{X_t\}_{t\in\mathbb{N}}$ forms an irreducible Markov chain as there is positive probability of eventually visiting any $X'$ starting from any $X$ since this is true for states $S'$ and $S$ in the original MDP (by irreducibility).

Let $\{B_t\}_{t\in\mathbb{N}}$ be the sequence of buffers that are observed. This also forms an irreducible Markov chain by the same reasoning as above since $\{X_t\}_{t\in\mathbb{N}}$ is irreducible. Additionally, the sequence of pairs $\{(X_{\text{BC}}^{(t)}, B_t)\}_{t\in\mathbb{N}}$ is an irreducible Markov chain.

Using the ergodic theorem (theorem 4.16 in [Levin and Peres, 2017]) on $\{(X_{\text{BC}}^{(t)}, B_t)\}_{t\in\mathbb{N}}$ with the projection function $f(x,y) = x$, we have that

$$
\lim_{T\to\infty} \frac{1}{T} \sum_{t=1}^{T} X_{\text{BC}}^{(t)} = \mathbb{E}\left[X_{\text{BC}}^{(t)}\right]
$$

where the expectation is over the joint stationary distribution of $(X_{\text{BC}}^{(t)}, B_t)$.

Using Lemma B.2 we can show that $\mathbb{E}\left[X_{\text{BC}}^{(t)}\right] = \mathbb{E}_\pi[\Delta]$, completing the proof.

$\square$

## B.5 Variance of BC-IR and IS

This lemma characterizes the variance of the BC-IR and IS estimators for a fixed buffer.

**Lemma B.3.** *Let $\mu_B = \mathbb{E}_\pi[\Delta|B]$ be the mean update on the batch $B$. Denoting the size of the buffer by $n$ and the number of size of the minibatch by $k$, let $X_{IS} = \frac{1}{k}\sum_{j=1}^{k}\rho_{z_j}\Delta_{z_j}$ (with each $z_j$ sampled uniformly from $\{1,...,n\}$) be the importance sampling estimator and $X_{BC} = \frac{1}{k}\sum_{j=1}^{n}\Delta_{i_j}$ (with each $i_j$ being sampled from $\ell \in \{1,...,n\}$ with probability proportional to $\rho_\ell$) be the bias-corrected importance resampling estimator. Then, the variances of the two estimators are given by*

$$\mathbb{V}(X_{\mathrm{IS}} \mid B) = \frac{1}{k}\left(\frac{1}{n}\sum_{j=1}^{n}\rho_j^2\|\Delta_j\|_2^2 - \mu_B^\top\mu_B\right)$$

$$\mathbb{V}(X_{\mathrm{BC}} \mid B) = \frac{1}{k}\left(\frac{\bar{\rho}}{n}\sum_{j=1}^{n}\rho_j\|\Delta_j\|_2^2 - \mu_B^\top\mu_B\right)$$

*Proof.* Since we condition on the buffer $B$, the only source of randomness is the sampling mechanism. Each index is sampled independently so we have that,

$$\mathbb{V}(X_{\mathrm{BC}} \mid B) = \frac{1}{k^2}\sum_{j=1}^{k}\mathbb{V}(\bar{\rho}\Delta_{i_j} \mid B) = \frac{1}{k}\mathbb{V}(\bar{\rho}\Delta_{i_1} \mid B)$$

and similarly $\mathbb{V}(X_{\mathrm{IS}} \mid B) = \frac{1}{k}\mathbb{V}(\rho_{z_1}\Delta_{z_1}|B)$

We can further simplify these expressions. For the IS estimator

$$\begin{aligned}
&\mathbb{V}(\rho_{z_1}\Delta_{z_1}|B) \\
&= \mathbb{E}[\rho_{z_1}^2\Delta_{z_1}^\top\Delta_{z_1}|B] - \mathbb{E}[\rho_{z_1}\Delta_{z_1}|B]^\top\mathbb{E}[\rho_{z_1}\Delta_{z_1}|B] \quad \text{by definition of } \mathbb{V}(\cdot) \\
&= \mathbb{E}[\rho_{z_1}^2\|\Delta_{z_1}\|_2^2|B] - \mu_B^\top\mu_B \quad \text{since } \rho_{z_1}\Delta_{z_1}|B \text{ is unbiased for } \mu_B \\
&= \frac{1}{n}\sum_{j=1}^{n}\rho_j^2\|\Delta_j\|_2^2 - \mu_B^\top\mu_B
\end{aligned}$$

The last line follows from the uniform sampling distribution. For the BC-IR estimator, recalling that $\bar{\rho} = \frac{1}{n}\sum_{i=1}^{n}\rho_i$, we follow similar steps,

$$\begin{aligned}
&\mathbb{V}(\bar{\rho}\Delta_{i_1}|B) \\
&= \mathbb{E}\left[\bar{\rho}^2\Delta_{i_1}^\top\Delta_{i_1}|B\right] - \mathbb{E}[\bar{\rho}\Delta_{i_1}|B]^\top\mathbb{E}[\bar{\rho}\Delta_{i_1}|B] \\
&= \mathbb{E}\left[\bar{\rho}^2\|\Delta_{i_j}\|_2^2|B\right] - \mu_B^\top\mu_B \quad \text{since } \bar{\rho}\Delta_{i_1}|B \text{ is unbiased for } \mu_B \\
&= \sum_{j=1}^{n}\bar{\rho}^2\frac{\rho_j}{\sum_{i=1}^{n}\rho_i}\|\Delta_j\|_2^2 - \mu_B^\top\mu_B \\
&= \frac{\bar{\rho}}{n}\sum_{j=1}^{n}\rho_j\|\Delta_j\|_2^2 - \mu_B^\top\mu_B
\end{aligned}$$

The fourth line follows from the sampling distribution of the $i_j$. $\qquad\square$

The following two theorems present certain conditions when the BC-IR estimator would have lower variance than the IS estimator.

**Theorem 3.3** Assume that $\|\Delta_j\|_2^2 > \frac{c}{\rho_j}$ for samples where $\rho_j \geq \bar{\rho}$, and that $\|\Delta_j\|_2^2 < \frac{c}{\rho_j}$ for samples where $\rho_j < \bar{\rho}$, for some $c > 0$. Then the BC-IR estimator has lower variance than the IS estimator.

*Proof.* We show $\mathbb{V}(X_{\text{IS}}|B) - \mathbb{V}(X_{\text{BC}}|B) > 0$:

$$\mathbb{V}(X_{\text{IS}}|B) - \mathbb{V}(X_{\text{BC}}|B) = \frac{1}{nk}\sum_{j=1}^{n}\|\Delta_j\|_2^2\,(\rho_j^2 - \bar{\rho}\rho_j)$$

$$= \frac{1}{nk}\sum_{s:\rho_s<\bar{\rho}}\underbrace{\|\Delta_s\|_2^2\,\rho_s}_{\leq c/\rho_s}\underbrace{(\rho_s - \bar{\rho})}_{\leq 0} + \frac{1}{nk}\sum_{l:\rho_l\geq\bar{\rho}}\underbrace{\|\Delta_l\|_2^2\,\rho_l}_{>c/\rho_s}\underbrace{(\rho_l - \bar{\rho})}_{\geq 0}$$

$$> \frac{1}{nk}\sum_{s:\rho_s<\bar{\rho}}\frac{c}{\rho_s}\rho_s\,(\rho_s - \bar{\rho}) + \frac{1}{nk}\sum_{l:\rho_l\geq\bar{\rho}}\frac{c}{\rho_l}\rho_l\,(\rho_l - \bar{\rho})$$

$$= \frac{c}{nk}\sum_{j=1}^{n}(\rho_j - \bar{\rho}) = 0$$

$\square$

**Theorem 3.4** Assume $\rho$ and the magnitude of the update $\|\Delta\|_2^2$ are independent

$$\mathbb{E}[\rho_j\|\Delta_j\|_2^2 \mid B] = \mathbb{E}[\rho_j \mid B]\,\mathbb{E}[\|\Delta_j\|_2^2 \mid B]$$

Then the BC-IR estimator will have equal or lower variance than the IS estimator.

*Proof.* Because of the condition, we can further simplify the variance equations from Lemma B.3. Let $c = \mathbb{E}[\|\Delta_j\|_2^2 \mid B]$. Then for BC-IR we have

$$\frac{\bar{\rho}}{nk}\sum_{j=1}^{n}\rho_j\,\|\Delta_j\|^2 = \frac{1}{k}\bar{\rho}\mathbb{E}\left[\rho_j\,\|\Delta_j\|^2 \mid B\right] = \frac{1}{k}\bar{\rho}\bar{\rho}c = \frac{1}{k}\bar{\rho}^2 c$$

and for IS we have

$$\frac{1}{nk}\sum_{j=1}^{n}\rho_j^2\|\Delta_j\|_2^2 = \frac{1}{k}\mathbb{E}\left[\rho_j^2\,\|\Delta_j\|_2^2 \mid B\right] = \frac{c}{k}\mathbb{E}\left[\rho_j^2|B\right]$$

Now when we take the difference, we get

$$\mathbb{V}(X_{\text{IS}}|B) - \mathbb{V}(X_{\text{BC}}|B) = \frac{c}{k}(\mathbb{E}\left[\rho_j^2|B\right] - \bar{\rho}^2)$$

$$= \frac{c}{k}\hat{\sigma}^2(\rho_j \mid B)$$

where $\hat{\sigma}^2(\rho_j)$ is the sample variance of the importance weights $\{\rho_j\}_{j=1}^{n}$ for $B$. Because the sample variance is greater than zero and $c \geq 0$, the BC-IR estimator will have equal or lower variance than the IS estimator.

$\square$

## B.6 Variance of BC-IR and WIS for a fixed dataset

The variance of BC-IR as compared to IS discussed in section 3.3 is only one comparison we can make. Similarly to bias, we can characterize the variance of the IR estimator relative to WIS-Optimal. $X_{\text{WIS}^*}$ is able to use a batch update on all the data in the buffer, which should result in a low-variance estimate but is an unrealistic algorithm to use in practice. Instead, it provides a benchmark, where the goal is to obtain similar variance to $X_{\text{WIS}^*}$, but within realistic computational restrictions. Because of the relationship between IR and WIS, as used in Theorem 3.1, we can characterize the variance of $X_{\text{IR}}$ relative to $X_{\text{WIS}^*}$ using the law of total covariance:

$$\mathbb{V}(X_{\text{IR}}) = \mathbb{V}\left[\mathbb{E}[X_{\text{IR}}|B]\right] + \mathbb{E}\left[\mathbb{V}[X_{\text{IR}}|B]\right]$$

$$= \mathbb{V}\left[X_{\text{WIS}^*}\right] + \mathbb{E}\left[\mathbb{V}[X_{\text{IR}}|B]\right]$$

where the variability is due to having randomly sampled buffers $B$ and random sampling from $B$. The second term corresponds to the noise introduced by sampling a mini-batch of $k$ transitions from the buffer $B$, instead of using the entire buffer like WIS. For more insight, we can expand

this second term, $\mathbb{E}\left[\mathbb{V}[X_{\mathrm{IR}}|B]\right] = \mathbb{E}\left[(\frac{1}{k}\sum_{j=1}^{k}\Delta_{i_j} - \frac{1}{n}\sum_{i=1}^{n}\Delta_i)^2|B\right]$, where we consider the variance independently for each element of $\Delta_i$ and so apply the square element-wise. The variability is not due to IS ratios, and instead arises from variability in the updates themselves. Therefore, the variance of IR corresponds to the variance of WIS, with some additional variance due to this variability around the average update in the buffer.

# C   Extended Experimental Results

## C.1   Markov Chain

This section contains the full set of markov chain experiments using several different policies. Results can be found in figure 4 and figure 6. See figure captions for more details.

Figure 5: Sensitivity curves for Markov Chain experiments with policy action probabilities [left, right] **left** $\mu = [0.5, 0.5], \pi = [0.1, 0.9]$; **center** $\mu = [0.9, 0.1], \pi = [0.1, 0.9]$; **right** $\mu = [0.99, 0.01], \pi = [0.01, 0.99]$.

Figure 6: Learning rate sensitivity plots for V-Trace (with the same settings as Figure 4). Three clipping parameters were chosen, including 1.0, 0.5 $\rho_{\max}$ and 0.9 $\rho_{\max}$, where $\rho_{\max}$ is the maximum possible IS ratio. For 1.0 $\rho_{\max}$, updates under V-trace become exactly equivalent to IS.

## C.2   Continuous Four Rooms

The continuous four rooms environment is an 11x11 2d continuous world with walls setup as the original four rooms environment grid world. The agent is a circle with radius 0.1, and the state consists of a continuous tuple containing the x and y coordinates of the agent's center point. The agent takes an action in one of the 4 cardinal directions moving $0.5 \pm \mathbb{U}(0.0, 0.1)$ in that directions and random drift in the orthogonal direction sampled from $\mathbb{N}(0.0, 0.01)$. The simulation takes 10 intermediary steps to more accurately detect collisions.

We use three behavior policies in our experiments. **Uniform:** the agent selects all actions uniformly, **State Variant:** the agent selects all actions uniformly except in pre-determined subsections of the environment where the agent will take down with likelihood 0.1 and the rest distributed evenly over the other actions, **State Weight Variant:** the agent selects all actions uniformly except in pre-determined subsections where the pmf is defined randomly. We also use two target policies. **Persistent Down**: where the agent always takes the down action, **Favored Down:** where the agent takes the down action with likelihood 0.9 and uniformly among the other actions with likelihood 0.1. We use a cumulant function which indicates collision with a wall and a termination function

Figure 7: SGD: Target Policy: **Top**: persistent down, **Bottom** favored down. Behaviour Policy: **left** State Variant **center** State Weight Variant **right** Uniform. Sample efficiency plots.

Figure 8: RMSProp Target Policy: **Top**: persistent down, **Bottom** favored down. Behaviour Policy: **left** State Variant **center** State Weight Variant **right** Uniform. Sample efficiency plots.

Figure 9: Incremental Experiments Target Policy: **Top**: persistent down, **Bottom** favored down. Behaviour Policy: **left** State Variant **center** State Weight Variant **right** Uniform. Sample efficiency plots.

Figure 10: SGD: Target Policy: **Top**: persistent down, **Bottom** favored down. Behaviour Policy: **left** State Variant **center** State Weight Variant **right** Uniform. Learning Rate Sensitivity

which terminates on collision and is 0.9 otherwise for all value functions. We present results using SGD and RMSProp over many algorithms and parameter settings in figures 7, 8, , and 10.