[Reviews · NeurIPS 2019]

Reviewer 1



Quality - The theoretical part is thorough enough, both in terms of the scope of results presented and quality of proofs (in the appendix). The empirical part is acceptable, but I feel could be improved (detailed below). Clarity - Apart from some minor peeves (detailed below), the paper is clear. Originality + Significance - The method introduced by the authors seems like a better approach to use IS weights, so while they did not introduce a new concept (I think most people who worked with importance sampling will recognize the methods as variants of the same idea), I think it is important to rethink how to use the idea of IS weights in ways which avoid their huge variance. Post rebuttal update - Despite the authors response, I think giving at least one simple model based baseline would be useful, as these make up an important approach to OPE and I don't think an IS paper is truly complete without such comparison, especially as these methods often perform better in practice. Nevertheless, I will keep my score of 7. A concern which arose during the reviewers' discussion regarded the validity of assumption 1. Especially in the regime of limited data, I believe that assuming transition are drawn i.i.d. is unlikely. While this assumption is used frequently in OPE, quite often for other estimators entire trajectories are sampled rather that individual transitions. Thus the trajectories are more likely to be i.i.d as opposed to individual trajectories. While I don't think the problematic nature of assumption 1 is big enough to prevent the paper from being published, if the paper gets accepted I strongly encourage the authors to have a thorough discussion in the text regarding when this assumption might hold or break.

Reviewer 2



The paper provides with a method for off-policy prediction in RL, where instead of using Importance Sampling weights in order to learn the value of one policy given samples from another policy, the authors propose a resampling scheme, where first a sub-sample of experiences from an experience buffer are sampled according to these importance sampling weights, and then an update is done without multiplying with the importance weight ratio itself. This allows reducing the variance since these ratios might have extreme values in case the behavior and target policy are very different. I found the paper interesting, however, I am not convinced how this differs from the Sampling Importance Resampling technique that is mentioned in the paper, which seems to already perform the same idea. In addition, I found some of the assumptions used in theorems quite restrictive and it is unclear to me why should they hold (see general comments below for details). The paper is generally well-written and clear, although some explanations are missing in my view, specifically in discussing the assumption used in various places, and in the experiments section where many details are missing. In summary, this is an interesting technique that seem to perform well in practice and have benefits over existing methods, but I feel the presentation should be improved and some additional explanations should be added. General Comments: - It is unclear to me how Importance Resampling (IR) is different from the Sampling Importance Resampling (SIR) method. You should explain this point. - You compare the bias of IR and BC-IR to the bias of WIS, but in the variance you compare to IS and not to WIS. why is this the case? why not compare to WIS? - line 126 - "For reasonably large buffers, \bar{\rho} will be close to 1" - why is this true?? this is unclear. - Assumption 1 is quite restrictive... assuming independence between transition tuples is not realistic since all transitions come from a behavior policy in the MDP, thus transitions are dependent in the sense that they come from a markov process - it is unclear why this assumption holds. - The conditions in Theorem 3.3 are quite restrictive, and the relaxation you propose still doesn't have to hold. Is there a reason for the conditions required for the low variance to hold in practice? also, you write once median and once mean, what is correct (line 189)? - line 203 - your last statement leaves the equation without giving the actual result - what is the expected difference between the two variances, where expectation is over buffers? - In the experiment section, you choose a target policy that chooses in a deterministic way the down action, which leads to extreme ratios of importance sampling. What about a softer comparison, when the target policy is not as extreme and still has some probability to perform other actions? - Figure 1 is very cluttered and it is hard to see where each algorithm is. Maybe consider enlarging it or somehow make it clearer. - In the experiments discussing the variance of the updates, you say that other policies are found in the appendix. Can you write whether the results are qualitatively similar for these other policies or are they different from the ones you present? - line 287, line 291 - you are referring to Figure 5- there is no figure 5 in the main text. Is this supposed to be figure 3? if so, what is a and b (in line 291)? - Figure 3 - what is the error specified of on the y-axis? how is the sensitivity to learning rate measured here? - Figure 3 - rightmost graph - what is the iteration number on the x-axis? It is not specified and so it is hard to see when does IS and IR converge to the same variance. - Car simulator experiment - you show the learning rate sensitivity, but not the convergence curve- which is the more important plot to display. Also, in figure 4, what is measured on the y-axis? how is this measured? It is unclear to me what are we seeing in this figure. It seems that the lowest error for IR is when the learning rate is higher, not lower as written in the text. It also seems that IR is either worse or better than IS, depending on the learning rate, i.e. it is not consistent. - line 319 - "we showed that IR is consistent" - but you only showed that BC-IR is consistent, not that IR is consistent. Technical Comments: - line 189 - "meanx" should be "mean" - Figure 2 - what is MARE on the y-axis? should it be MAVE? --- Added after author feedback --- I have read the author feedback. My two main concerns regarding this work were the novelty of it and the usage of assumption 1. Regarding the novelty - the author mention in their feedback that the novelty is by applying the method (which is not different from SIR) to the RL setting where the buffer is not fixed but online. However, once they assume assumption 1 and assume in all analysis that the buffer is a sample of n i.i.d samples from some fixed distribution, then this is not a valid argument... it does not capture the online nature of RL and the online nature of the content of the buffer. Thus, this becomes similar to previous works using this in domains other than RL. I am also not quite convinced that the assumptions in theorem 3.3 are easily met, and the explanation given in the feedback isn't very formal but more 'intuitive', which weakens the result. My second main concern was the use of assumption 1, however, since it seems other reviewers advocate that this assumption is common, and given that the authors will also clarify and revise the experimental section, I am ready to increase my score.

Reviewer 3



Thank you for addressing my comments. In particular Q3 was addressed and I recommend (1) making the explanation re relationship between learning rate and variance in updates obvious; and (2) mentioning that error bars are in the caption for Figure 3. Given that two reviewers were confused about C1f I recommend explaining it in a footnote. I will maintain my score after taking into account that I might have been over enthusiastic about the theoretical contribution. Thanks! ========= Originality: Though there exist other works that weight samples from a replay buffer this paper presents a concrete alternative to one of off-policy learnings biggest woes: importance sampling. The paper provides extensive theoretical treatment and empirical validation that are original as far as i know. Here are few ways to make the originality clear: O1: Some discussion of other approaches to non-uniform sampling of the replay buffer and how your technique varies (my understanding is that you are doing GVF learning compared to something like Q-learning which does not require the sampling distribution to match the policy distribution) O2: some discussion, perhaps in the conclusion, about extensions to control would be useful. Quality: The theoretical contributions are of high quality. There are a few clarity issues i found which are specified in the next (clarity) section. I especially liked the detailed experimental setup and the motivations for using each of the domains. I was confused about one thing: Q1: What Is the difference between “Step” and “Number of updates” in Figure 1? I think this should be clarified. I think the experiments have adequately verified the improved sample efficiency and reduced variance claims. However, I find that some of the conclusions might be better supported by a more in-depth discussion of the results. I outline these below: Q2a: L238-249: Though this paragraph makes sense, it can be improved significantly if the figures are referred to more often in the text. (For example L240-243 which figure? Which part of it?) Q2b: I think L287 is referring to the wrong figure. Q3: L290-292: I think the kind of variance referred to here is different from what we care about. In particular, I think it means “variance in random restarts” and not necessarily “variance in the updates” (as shown in Fig 3 right). I think its fine if this was clarified. However, I should point out that Fig 3 left and center do not have error bars so we cant really say anything about “variance in random restarts” but only regarding “learning rate sensitivity” Clarity The paper is overall well motivated and written. There are a few issues I have with the clarity of the mathematic claims and proofs: C1a: The authors refer to the average update direction as X. This seems to be also mixed with the transition tuple. I strongly recommend decoupling the two. This is particularly problematic in Theorem 3.2 where X refers to both transitions in the buffer as well as the importance (re)sampling updates. C1b: There are some steps of the proofs in Appendix B that are somewhat confusing. Though i was able to follow most of it and didn’t find any obvious errors, I believe this paper could almost serve as a cornerstone paper if each of the steps are made explicit. For example, I don’t know how the equations after L500 progress. Additionally, it seems that mu_B refers to two different quantities in this proof and might cause unnecessary confusion. C1c: L514 missing a >= 0 on the second line. C1d: Theorem 3.4 please restate the math statement after the text statement in L196 similar to L185. C1e: L169-170: “For high magnitude ratios…” I am not sure why this sentence is here. It sounds important but how does the work get around this issue. Is L171-173 an empirical observation about it? C1f: L126-127: I understand that for small buffers rho \approx 1 and therefore \bar{\rho} \approx 1 but why is it true for large buffers? Additionally, there are certain parts of the text that were confusing to me: C2a: Minor typo L189 “meanx” C2b: L193 “and so potentially...” is incorrectly worded. C2c: L100-101: why is it not straightforward? Perhaps you mean to say, “Since it is not straightforward…” C2d: L53-55 I have no idea what this sentence is trying to say. C2e: I think L60-62 should come earlier in the text as a very clear motivating reason for not wanting to use IS. C2f: L284 confusing as to what this means “simulating the many possible updates …” I personally enjoyed the colour consistency and the clarify of the figures. There were a few figures that could use some clarification: C3a: Figure 7 and 8 should have “sample efficiency plots” come to the start of the figure. C3b: Figure 4: are the arrows for BCIR in the wrong place? C3c: Is it worth converting the exponents in the log scale plot to integers?

[Author Response · NeurIPS 2019]

Thank you for your in-depth and constructive reviews, they will give us an excellent chance to further improve the paper. We address the reviewers concerns individually below. We will not address all the typos and editing catches, but will fix all of these in the final draft.

**Reviewer 1:**

- The goal here is to compare model-free methods, augmented with a buffer. A careful comparison between model-free and model-based approaches for OPE would be interesting and extremely valuable for the community, but is beyond the scope of this paper.

- It is a good suggestion to provide intuition for the proofs. We will include such a discussion in the camera-ready, which allows for an additional page.

Minor Concerns: We will address these concerns in the revision. We will improve notation consistency and clarity and include diagrams of the maze in the appendix.

**Reviewer 2:**

- SIR is a general strategy; in fact, there are a number of similar approaches with different names [Smith and Gelfand, 1992]. The main novelty here is investigating its use in RL, where the online setting requires us to consider a moving window dataset—rather than a fixed batch.

- The theoretical comparison of the bias of IR and WIS-Optimal is natural, because we show they are equal. IS is unbiased, so that comparison is not interesting. In practice, though, we cannot actually use WIS-Optimal, as it is a full batch approach. Empirically, then, it makes sense to compare to other mini-batch methods, like IS. We did not compare to WIS-minibatch due to the poor empirical performance, likely due to the additional bias of that estimator.

- $\bar{\rho} \approx \mathbb{E}[\rho(a|s)] = \mathbb{E}[\frac{\pi(a|s)}{\mu(a|s)}] = \sum_{s,a} \frac{\pi(a|s)}{\mu(a|s)} \mu(a|s) d_{\mu}(s) = 1$.

- Assumption 1 is common for analyzing OPE estimators. The idea is that we are effectively sampling from the stationary distribution, even though we know we are in Markov settings. An important next step is to consider alternative noise assumptions in sampled data.

- The result in line 203 is that we can directly use the prior results to look at the expected difference in variances over many buffers (i.e. these statements say our result holds across buffers of smaller sizes).

- Because $\bar{\rho} \approx 1$. When $\rho$ is lower than the average it will make the rhs a large number, but when $\rho$ is greater than the average we expect it to lower the rhs of the equation. As learning progresses, we expect the samples w/ high $\rho$ to learn more quickly (thus having lower error). 'mean' is the correct one.

- We also looked at "softer" target policies, where similar conclusions can be drawn (see appendix). All the results presented in the appendix are qualitatively similar.

- The parameter sensitivity provides more information, because it gives some sense of how these might perform in practice for realistically chosen parameters, rather than optimal parameters. We will include the learning curves in the appendix for completeness.

- MARE is Mean Absolute Return Error. We use MARE when it is not tractable to compute the value function using dynamic programming or analytically (and otherwise MAVE).

**Reviewer 3:**

For O1, you are correct in your understanding. We will use some of the additional space in the camera-ready to include a brief discussion on O1 and O2.

Q1: The Steps corresponds to the Number of Interactions with the environment. The agent can update more or less frequently than every step. The Number of Updates for Figure 1(a) is once every 16 Steps. We also show performance for different Update frequencies in Figure 1(b).

Q3: Because the experiment is run 100 times, like in all the plots in figure 3, the error bars are not visible. The parameter sensitivity plots could provide some information about variance of the updates. If the variance of the updates is higher, we expect the magnitude of the largest updates to also be higher. This means a lower step size is needed to prevent divergence. A wider trough of the sensitivity curve could reflect lower variance in the updates, though as acknowledged in the paper, this is very much a proxy and we cannot make any strong conclusions based on it.

C1(f): See point 3 for Reviewer 2.

Q2, C1(a-e), C2, and C3: we will take all these points into consideration, and work to maximize clarity in the final revision. L287 - You are correct, this should be figure 3.

[Meta-Review · NeurIPS 2019]

Although the reviewers like the paper, they are all concerned about the novelty of the results. The author admit in their response that the novelty is not in the estimator, IR or SIR, but in its application to RL. But then the main assumption in the analysis of the paper (Assumption 1) says that the samples in the buffer are i.i.d., which is not a valid assumption in RL. However, after all, we think introducing IR or SIR and using it in the problem of off-policy evaluation is important and beneficial to the community. I strongly recommend that the authors address the issues raised by the reviewers and clarify the contribution of the work (that Assumption 1 is in contrast with the online nature of RL) in the final version of the paper.